# INDICVOICES-R: Unlocking a Massive Multilingual Multi-speaker Speech Corpus for Scaling Indian TTS

**Ashwin Sankar**[*]    **Srija Anand**[*]    **Praveen Srinivasa Varadhan**[*]    **Sherry Thomas**

**Mehak Singal**    **Shridhar Kumar**    **Deovrat Mehendale**    **Aditi Krishana**    **Giri Raju**

**Mitesh Khapra**[†]

AI4Bharat, Department of Computer Science & Engineering
Indian Institute of Technology Madras
Chennai, Tamil Nadu, India - 600036

## Abstract

Recent advancements in text-to-speech (TTS) synthesis show that large-scale models trained with extensive web data produce highly natural-sounding output. However, such data is scarce for Indian languages due to the lack of high-quality, manually subtitled data on platforms like LibriVox or YouTube. To address this gap, we enhance existing large-scale ASR datasets containing natural conversations collected in low-quality environments to generate high-quality TTS training data. Our pipeline leverages the cross-lingual generalization of denoising and speech enhancement models trained on English and applied to Indian languages. This results in IndicVoices-R (IV-R), the largest multilingual Indian TTS dataset derived from an ASR dataset, with 1,704 hours of high-quality speech from 10,496 speakers across 22 Indian languages. IV-R matches the quality of gold-standard TTS datasets like LJSpeech, LibriTTS, and IndicTTS. We also introduce the IV-R Benchmark, the first to assess zero-shot, few-shot, and many-shot speaker generalization capabilities of TTS models on Indian voices, ensuring diversity in age, gender, and style. We demonstrate that fine-tuning an English pre-trained model on a combined dataset of high-quality IndicTTS and our IV-R dataset results in better zero-shot speaker generalization compared to fine-tuning on the IndicTTS dataset alone. Further, our evaluation reveals limited zero-shot generalization for Indian voices in TTS models trained on prior datasets, which we improve by fine-tuning the model on our data containing diverse set of speakers across language families. We open-source code and data[3] for all 22 official Indian languages.

## 1   Introduction

Scaling training data has been crucial in achieving better zero-shot speaker and style generalization for English Text-To-Speech (TTS) systems, with studies demonstrating the benefits of increasing data from a few hundred hours to beyond 10,000 hours [1, 2, 3, 4, 5]. Indeed, state-of-the-art models like Natural Speech 3 [3] are trained on 200,000 hours of data. Additionally, using conversational

---

[*] Equal Contribution

[†] Corresponding Author: miteshk@cse.iitm.ac.in

[3] **Dataset Link -**`https://github.com/AI4Bharat/IndicVoices-R`

or extempore speech has been shown to enhance the naturalness of TTS systems compared to read-speech data. However, scaling training for Indian TTS has been challenging due to limited resources. Existing TTS datasets for Indian languages collectively cover only up to 14 of the 22 scheduled languages of India, with 1-2 speakers per language and primarily consisting of read-speech recordings which are not rich in prosody and expression. These limitations significantly hinder the progress of Indian TTS towards natural-sounding speech.

One common approach to scale the training data is to mine TTS data from online public sources such as YouTube. However given the low resource status of Indian languages, finding manually transcribed data for all 22 Indian languages on the internet are challenging, and the available data often lacks professional studio quality. Further, such in-the-wild speech data typically contains overlapping speakers, music, and a low signal-to-noise ratio. Instead, we choose to repurpose existing ASR datasets for TTS, a popular approach that has unlocked better data in English, such as with LibriTTS-R [6] and LibriLight (60k hours) [7]. For doing so, several candidate Indian ASR datasets exist, including KathBath[8], Shrutilipi [9], FLEURS [10], and IndicVoices[11]. Among these, IndicVoices is the most promising due to its unique characteristics: it covers all 22 scheduled languages of India, includes both read-speech and conversational formats, and involves a large number of speakers. These characteristics are also desirable in a TTS dataset, enabling the creation of human-like speech across all scheduled Indian languages while achieving zero-shot speaker generalization.

We propose a pipeline to denoise and enhance ASR data, effectively addressing the poor quality of audio samples, which often include background noise, intermittent chatter, echoes, and high reverb. Such data is frequently collected at a lower sampling rate of 16 KHz, compared to the 48 KHz typically found in professional studio recordings. We leverage the significant progress in Speech Enhancement and Restoration models to effectively address these issues. We find that these models, although pre-trained on English, show good cross-lingual generalization and can enhance Indian speech data without affecting intelligibility. Using the proposed pipelined we build IndicVoices-R (IV-R), the largest multilingual Indian Text-to-Speech (TTS) dataset derived from IndicVoices, unlocking 1,704 hours of high-quality speech data from 10,496 speakers across 22 Indian

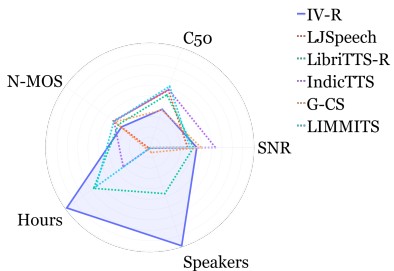

Figure 1: IndicVoices-R (IV-R) surpasses existing TTS datasets in terms of sheer volume of hours and speakers while having comparable speech quality to such TTS datasets, as assessed across a range of speech-quality metrics.

languages. As seen in Figure 1, our dataset matches the speech and sound quality of gold-standard TTS datasets like LJSpeech, LibriTTS, and IndicTTS, as measured by the NORESQA [12], SNR, and C50 scores [13]. We complement IV-R, by also releasing a comprehensive benchmark to test zero-shot, few-shot, and many-shot speaker generalization of Indian TTS systems.

Using IV-R, we enhance the zero-shot speaker generalization of an English pre-trained VoiceCraft model and also enables it to support multiple Indian speakers and languages. We show the former by contrasting its performance with a model fine-tuned on a less diverse Indian TTS dataset. We open-source code and data for all 22 official Indian languages.

## 2   Background

**Achieving Human-level Speech Quality** NaturalSpeech [14] is a pioneering work that achieved human-level speech quality in neural TTS with large-scale phoneme encoder pre-training and improved architectures leveraging fully end-to-end training. Similarly, StyleTTS2 [15] surpasses the quality of human recordings by using multiple techniques such as style diffusion and adversarial training with large speech language models such as WavLM [16]. However, a limitation of these models trained on professional studio-quality datasets is that they often contain only single-speaker data or cover limited speaking styles such as read-speech or narration, thereby limiting the ability of these TTS systems to generalize to a variety of voices, speaking styles, and prosodic variations. To overcome this barrier, several works [1, 3, 4, 5, 17] scale TTS training using up to 1B parameters on large-scale multi-speaker found data consisting upto 100K hours.

**Speech-Text Datasets** LJSpeech [18] is a foundational TTS dataset that releases 24 hours of single-speaker recordings from passages of non-fiction books. LibriTTS [19] releases multi-speaker data derived from LibriVox audiobooks, providing 585 hours of speech from 2,456 speakers. VCTK [20] offers another significant resource with approximately 44 hours of recordings from 109 native English speakers with various accents. The introduction of several multi-speaker datasets [19, 20, 21] has enabled the development of more prosodic TTS adaptive to multiple speaking styles and speaker voices. More recently, there has been a growing demand for large (un)labeled speech corpora [22, 23] for training TTS systems. Existing Indian speech-text datasets like IndicTTS [24], IndicSpeech [25], Google-CrowdSourced (Google-CS) [26] and LIMMITS [27] are either considerably smaller in scale, exhibit lower speaker or style diversity compared to open-source multilingual datasets. This limited scale restricts the development of robust and versatile TTS models for Indian languages. In this work, we take inspiration from LibriTTS-R [6] dataset, and attempt to enhance and restore IndicVoices [11] to bridge the data gap of large-scale diverse multi-speaker data for Indian languages.

**TTS for Indian Languages** Initial neural works [28, 29] that build TTS for Indian languages explore the importance of pooling speaker data across multiple languages within the same language family to overcome the data constraints and train good quality TTS systems. Subsequent work [30] outlines design choices for training Indian TTS and releasing better quality monolingual multi-speaker models for 13 Indian languages. More recently, work in [31] demonstrates that training multilingual, multi-speaker text-to-speech systems based on language families, like Indo-Aryan and Dravidian, can effectively leverage limited transcribed data and adapt to new languages in low-resource scenarios.

**TTS Benchmarks** Comparison of state-of-the-art TTS has often been challenging due to the lack of standardized benchmarks. Conventionally, a couple of works [2, 32, 33] evaluate on an agreed-upon set of 50 particularly hard sentences. Several works evaluate on the train-test splits of the original dataset they train on, especially for works training on LJSpeech and LibriTTS. EXPRESSO [34] offers 47 hours of North American English data from 4 speakers across 26 styles, providing a benchmark for TTS prosodic and speaker-style variation. BaseTTS [35] introduces benchmarks for questions, emotions, compound nouns, syntactic complexity, foreign words, punctuations, and paralinguistics. However, the Indian TTS community faces a significant gap in the availability of benchmarks for TTS evaluation. We aim to address this by releasing the first zero-shot, few-shot, and many-shot cross speaker benchmark, for all 22 Indian languages.

# 3 INDICVOICES-R

INDICVOICES-R is derived from IndicVoices, an existing large-scale multilingual multi-speaker speech corpus for Indian languages. This section explains our rationale behind choosing IndicVoices, and details our data pipeline to restore this dataset to TTS quality. We also compare our enhanced version, IndicVoices-R, with several existing TTS datasets and show that it achieves on-par quality. Finally, we provide detailed statistics and the format of our released dataset. IndicVoices-R, boasts several key features that make it an ideal dataset choice for scaling Indian TTS systems:

**(i) Comprehensive Coverage:** It is the first dataset to encompass all 22 Indian languages, offering between 9 to 175 hours of speech data per language.

**(ii) Speaker Diversity:** With a vast pool of over 10,496 speakers (greater than any existing TTS dataset) representing various demographics and linguistic backgrounds, it ensures rich diversity crucial for attaining good cross-speaker generalization in TTS.

**(iii) Natural Recordings:** The dataset predominantly consists of extempore recordings (93.25%), capturing spontaneous speech, which is pivotal for achieving naturalness in synthesized speech.

**(iv) High-Quality Samples:** The quality of the dataset's samples matches or exceeds that of several existing large-scale multi-speaker TTS datasets, underscoring its efficacy in TTS model training.

## 3.1 Choice of ASR Dataset

While several Automatic Speech Recognition (ASR) datasets like Kathbath [36], Shrutilipi [9], IndicVoices [11], and Dhwani [37] exist, IndicVoices emerges as the most suitable choice for constructing an Indian Text-to-Speech (TTS) dataset. Particularly, we would like to highlight several key advantages of IndicVoices based on (i) data quality, (ii) speaker and language coverage, (iii)

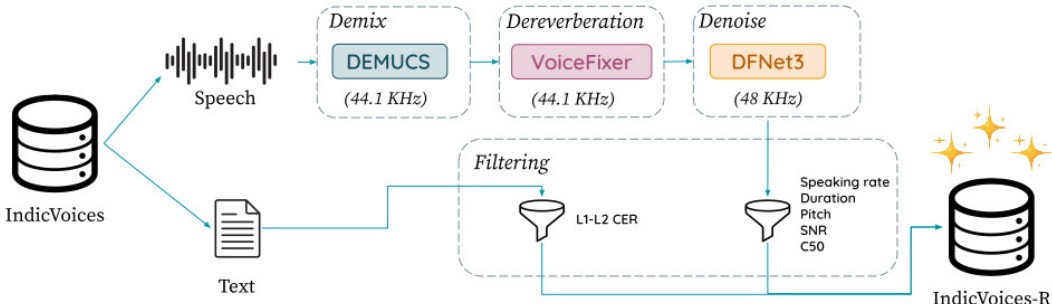

Figure 2: Data pipeline for restoring IndicVoices that demixes, dereverberates, and denoises speech samples, followed by filtering of speech-text pairs to yield an enhanced IndicVoices-R dataset.

diversity, and (iv) ethical considerations. Unlike the other datasets, which are limited to lower sampling rates (16 kHz), IndicVoices prioritizes high-fidelity recordings at 44.1 kHz. Furthermore, manual annotation with a stringent quality assurance process guarantees high-quality text alignments with audio samples. This surpasses the need for relying on ASR-based transcription that would potentially introduce errors when labelling other large-scale speech corpuses such as Dhwani. In terms of ethical considerations, IndicVoices adheres to data protection regulations and obtains informed consent from speakers. This commitment to responsible data collection practices starkly contrasts with approaches that rely on web-mined data, where it is more challenging to adhere to licensing terms. Finally, IndicVoices stands out as the only existing speech corpus that covers all 22 Indian languages, offering rich speaker diversity and a wide range of textual content across various domains and speaking styles. This makes it the ideal candidate for developing a comprehensive TTS dataset.

## 3.2 Data Pipeline

To enhance INDICVOICES-R, we employ a comprehensive data pipeline comprising several key steps which we illustrate in Figure 2.

**Step 1: Pre-processing audios** IndicVoices consists of recordings at 44.1kHz and 8kHz. We only consider audios at 44.1KHz consisting of extempore and read-speech, as we wish to derive a high-quality speech dataset. We also filter out all audio samples greater than 30s in duration. We upmix mono channels to stereo channels in all audios using ffmpeg [38].

**Step 2: Demixing ASR-quality audios** IndicVoices was recorded in natural environments, which means the audio samples often contain various types of background noise, intermittent or overlapping chatter, echo, and low volume levels. These issues can significantly impact the quality and usability of the dataset for Text-to-Speech (TTS) training. To address these issues, we utilize HTDemucs [39], a state-of-the-art deep learning model for audio source separation and noise reduction.

**Step 3: Dereverberation of Demixed audios** Despite the denoiser effectively eliminating most background noises from the audio samples, we encountered instances where the audios still exhibited high reverb and digital artifacts overlapping with the speech or vocals. These artifacts can degrade the overall quality and intelligibility of the audio. By running the entire denoised set through VoiceFixer [40], we observed remarkable improvements in reducing reverberation without any loss of intelligibility in the audio.

**Step 4: Speech Enhancement** We noticed that VoiceFixer introduced certain digital artifacts in samples. To filter these out, we rely on DeepFilterNet3 [41], which operates on full-band spectrograms, and found that it effectively eliminates these artifacts while preserving speech quality.

**Step 5: Filtering**

**Audio** We employ a thorough filtering process to ensure high audio quality based on speech quality metrics that we discuss further in Section 3.3. Audios must meet specific thresholds: $C50 \geq 30\,\mathrm{dB}$ for minimal reverberation, $SNR \geq 25\,\mathrm{dB}$ for clear speech, $0.2 < \text{duration} < 30$ seconds for appropriate length, utterance-pitch-mean $\leq 350\,\mathrm{Hz}$ for natural pitch, utterance-pitch-standard-deviation $\leq 150\,\mathrm{Hz}$ for pitch consistency, and speaking-rate $\leq 30$ characters per second for mainstream pace. This rigorous selection process ensures that our dataset consists of high-quality audio samples appro-

Table 1: A comparison of IndicVoices-R with other popular TTS datasets, both English datasets and Indic datasets.

| Dataset | # Uttr. | # Hours | # Spk. | # Lang. | N-MOS (↑) | SNR (↑) | C50 (↑) | F0 (↑) |
|---|---|---|---|---|---|---|---|---|
| LJSpeech | 13.1K | 24 | 1 | 1 | 4.36 | 58.03 | 58.56 | 208.04 |
| LibriTTS | 154.6K | 585 | 2456 | 1 | 3.76 | 59.60 | 57.33 | 159.51 |
| IndicTTS | 141.9K | 284 | 27 | 14 | 4.29 | 65.38 | 58.78 | 166.04 |
| Google-CS | 25.2K | 41 | 261 | 7 | 4.20 | 61.63 | 53.24 | 192.86 |
| LIMMITS | 246K | 560 | 14 | 7 | 4.30 | 58.69 | 59.60 | 176.37 |
| **Ours** | 689.6K | 1704 | 10496 | 22 | 3.38 | 60.47 | 53.45 | 178.91 |

Table 2: Distribution of number of speakers (S) and duration in hours (H) of our data across age groups and genders.

| Age | Male | | Female | |
|---|---|---|---|---|
| Group | # S | # H | # S | # H |
| 18-30 | 2109 | 341.69 | 2284 | 369.72 |
| 30-45 | 1414 | 234.87 | 1575 | 250.85 |
| 45-60 | 881 | 154.04 | 1015 | 162.06 |
| 60+ | 626 | 108.80 | 592 | 97.18 |
| Total | 5030 | 839.41 | 5466 | 879.80 |

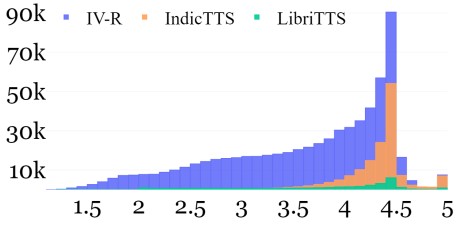

Figure 3: N-MOS of IV-R vs. existing TTS datasets.

priate for TTS training, free from irregularities that could impede the performance of TTS models trained on such data.

**Transcript** IndicVoices provides two levels of transcriptions tailored to diverse needs. In Level 1, transcriptions faithfully reflect spoken language, akin to verbatim transcripts. Level 2 adopts standardized word forms, appealing to both linguistic purists and everyday users. To uphold precision and clarity, we exclude utterances with a character error rate surpassing 5% between the two levels, ensuring that alignment between transcriptions and speech remains consistent and reliable.

**Step 6: Post-Processing** We randomly sampled audio files from the enhanced set and found that they differed in volume. To fix this, we employed the normalize function in PyDub [42] to adjust the volume of an audio segment and bring its peak amplitude close to the maximum possible level, while maintaining a specified headroom (set to 0.1 in dB) to avoid clipping. Just as we ensure consistency in audio volume through normalization, one might question the value of normalizing text. Since we use the verbatim version of the transcript, no additional normalization is required and we are able to maintain fidelity to a speaker's utterances, capturing nuances, colloquialisms, and idiosyncrasies inherent in natural language.

## 3.3 Comparison with Existing TTS Datasets

To prove the utility of our data pipeline in yielding high-quality samples, we compare INDICVOICES-R against popular TTS datasets including LJSpeech, LibriTTS, IndicTTS, Google-CS and LIMMITS. We attempt to show that the speech quality of our dataset is on par with other existing TTS datasets while surpassing them in terms of speaker diversity, vocabulary diversity, and corpus size.

### 3.3.1 Speech Quality

We report four metrics (i) NORESQA-MOS(N-MOS), (ii) Signal-to-Noise Ratio (SNR), (iii) C50 and (iv) utterance-level pitch mean ($F_0$) values to measure the quality and clarity of the speech samples. The NORESQA [12] framework uses a Non-Matching Reference (NMR) along with the given test speech signal to estimate speech quality. Particularly, we calculate the N-MOS score using randomly chosen samples from the LibriTTS set as the NMR. We used Brouhaha [13] to compute SNR and C50 and PENN [43] to compute utterance level pitch mean ($F_0$) using the Dataspeech [44] repository.

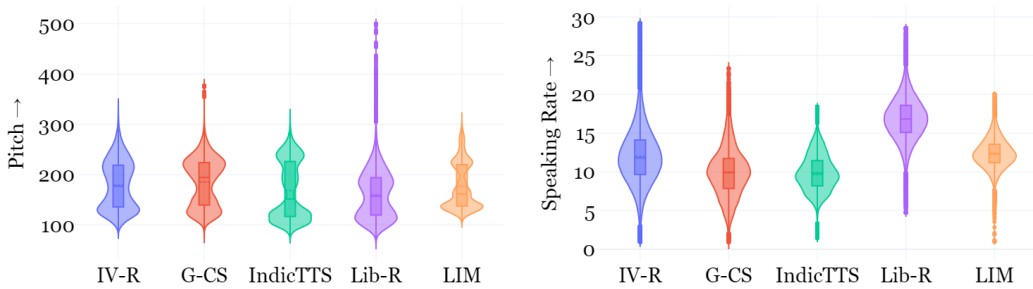

Figure 4: Comparison of Pitch and Speaking Rates of IV-R vs. existing TTS datasets.

In Table 1, we highlight that INDICVOICES-R surpasses all other datasets in terms of sheer volume of number of utterances, hours, speakers, and languages. Furthermore, while studio-quality TTS datasets such as LJSpeech, IndicTTS, and LIMMITS achieve higher N-MOS scores than ASR-purposed datasets like LibriTTS and IndicVoices-R. Nevertheless, we plot the distribution of N-MOS (Figure 3) across datasets and find that INDICVOICES-R has more utterances than IndicTTS and LibriTTS with N-MOS scores greater than 4.

Our dataset also maintains competitive noise levels (measured by SNR) with other datasets, indicating its suitability for TTS. Its C50 score of $53.45\,\mathrm{dB}$, is only slightly lower than studio-quality datasets (LJSpeech, IndicTTS, LIMMITS), indicating good speech clarity. In contrast, studio-quality datasets like LJSpeech, IndicTTS, and LIMMITS have higher mean C50 values, reflecting the superiority of recording in studio environments. The $F_0$ values, representing pitch, are within a comparable range across other Indian TTS datasets, indicating expected pitch levels. Overall, INDICVOICES-R, with its extensive linguistic diversity and considerable dataset size, offers a balance between quality and quantity, making it a valuable dataset for building TTS systems.

### 3.3.2  Speech Diversity

**Speaker Diversity** Covering as many as 10,469 speakers across the 22 official Indian languages, INDICVOICES-R has $40\times$ more speakers than the previous Indian TTS corpora with the highest speaker diversity, Google-CS. Table 2 elucidates the diversity and balance in the distribution of the number of speakers and data hours in each age group. This comprehensive diversity is pivotal for training TTS systems capable of zero-shot speaker and style adaptation.

**Style Diversity**: The analysis of pitch distribution and speaking rates across various datasets, as depicted in Figure 4, underscores the inherent naturalness of our INDICVOICES-R. Our dataset exhibits a rich array of mean pitch values, extending to 350 Hz, contrasting with the narrower pitch range observed in datasets like IndicTTS, which predominantly comprises meticulously recorded read-speech in controlled settings. Furthermore, our dataset demonstrates a mean speaking rate of 12 words per second, with a notable dispersion, in contrast to the more uniform rate of 9.88 words per second found in IndicTTS, indicating a prevalence of spontaneous speech and naturalness.

**Vocabulary Diversity** We compare character bigram for INDICVOICES-R and IndicTTS in Figure 5 and found our data to be marginally better in 8 languages while comparable in the remaining 5 languages that IndicTTS covers.

### 3.4  Data Statistics and Format

We present comprehensive statistics of INDICVOICES-R in Table 3. Notably, our dataset is the first to publicly open-source TTS data for 9 Indian languages - Dogri, Kashmiri, Konkani, Maithili, Nepali, Sanskrit, Santali, Sindhi, and Urdu. Overall, our dataset consists of 1,704 hours of high-quality speech data from 10,496 speakers across 22 Indian languages with over 1M unique words and 690K utterances. We release the metadata files in JSONL format and explain the attributes provided for each speech-text pair in Appendix.

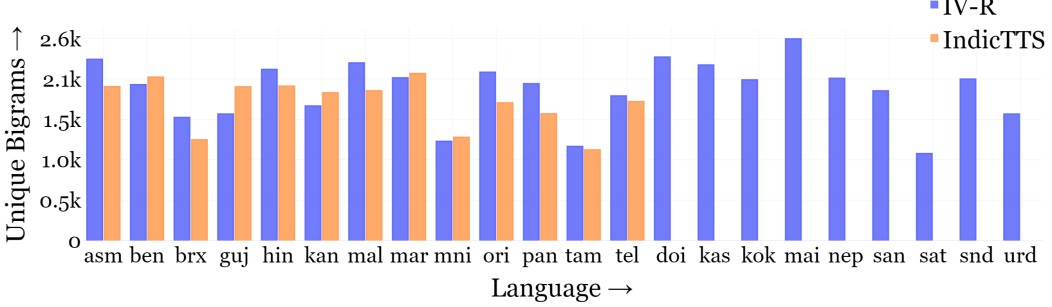

Figure 5: Unique character bigram counts across languages for our dataset and IndicTTS. Languages are reported using ISO-639-3 language codes.

Table 3: Detailed Statistics of INDICVOICES-R across languages

| Languages | # Hours | | | # Utterances | | # Speakers | | # Words | # Bi-grams |
|---|---|---|---|---|---|---|---|---|---|
| | Read | Extempore | Total | Count | Avg. (s) | Count | Avg. (s) | | |
| **Assamese** | 7.05 | 168.29 | 175.34 | 73077 | 8.64 | 928 | 680.43 | 73846 | 2302 |
| Bengali | 4.63 | 107.37 | 111.99 | 40943 | 9.85 | 617 | 654.20 | 45095 | 1979 |
| Bodo | 2.17 | 169.89 | 172.05 | 83976 | 7.38 | 941 | 658.90 | 97526 | 1563 |
| Dogri * | 3.64 | 67.04 | 70.68 | 27967 | 9.10 | 470 | 541.16 | 33127 | 2330 |
| Gujarati | 1.06 | 7.88 | 8.94 | 3304 | 9.74 | 45 | 707.91 | 11773 | 1607 |
| Hindi | 4.31 | 70.28 | 74.60 | 27557 | 9.75 | 399 | 672.83 | 24697 | 2173 |
| Kannada | 3.87 | 40.74 | 44.61 | 18127 | 8.86 | 452 | 354.56 | 53683 | 1708 |
| Kashmiri * | 7.12 | 57.87 | 64.99 | 26134 | 8.95 | 450 | 517.89 | 50424 | 2230 |
| Konkani * | 5.87 | 47.19 | 53.06 | 22357 | 8.54 | 228 | 839.34 | 47467 | 2041 |
| Maithili * | 6.18 | 75.59 | 81.77 | 32483 | 9.06 | 627 | 462.78 | 50560 | 2561 |
| Malayalam | 5.47 | 77.10 | 82.57 | 32544 | 9.13 | 462 | 641.73 | 90052 | 2256 |
| Manipuri | 0.77 | 23.22 | 23.99 | 9312 | 9.28 | 127 | 664.05 | 32352 | 1259 |
| Marathi | 4.80 | 46.10 | 50.90 | 20164 | 9.09 | 359 | 507.50 | 43059 | 2066 |
| Nepali * | 8.89 | 96.99 | 105.87 | 43545 | 8.75 | 716 | 533.93 | 58952 | 2060 |
| Odia | 6.04 | 64.90 | 70.95 | 26450 | 9.66 | 441 | 579.28 | 37601 | 2138 |
| Punjabi | 6.20 | 68.74 | 74.94 | 27788 | 9.71 | 335 | 805.40 | 25910 | 1992 |
| Sanskrit * | 4.82 | 30.93 | 35.75 | 14604 | 8.81 | 161 | 787.51 | 35271 | 1901 |
| Santali * | 6.31 | 70.06 | 76.37 | 35155 | 7.82 | 309 | 890.28 | 33102 | 1104 |
| Sindhi * | 2.50 | 7.98 | 10.48 | 4197 | 8.99 | 204 | 191.67 | 11900 | 2050 |
| Tamil | 11.94 | 87.53 | 99.47 | 40464 | 8.85 | 1084 | 328.37 | 96167 | 1195 |
| Telugu | 6.91 | 129.49 | 136.40 | 48485 | 10.13 | 681 | 721.46 | 90301 | 1837 |
| Urdu * | 4.45 | 74.17 | 78.61 | 30935 | 9.15 | 460 | 624.67 | 23752 | 1607 |
| **Total** | 114.98 | 1589.36 | **1704.34** | 689568 | - | **10496** | - | 1066617 | 41959 |

# 4 IndicVoices-R Benchmark

We complement INDICVOICES-R by also providing a train set and a carefully crafted held-out set to truly test the zero-shot, few-shot, and many-shot speaker generalization capabilities of TTS systems for Indian speakers and languages. We call this benchmark as INDICVOICES-R Benchmark and illustrate its statistics in Table 4. To construct this benchmark, we attempt to maximize the number of zero-shot and few-shot speakers across both genders and age groups while also ensuring we do not lose out on too much training data. We select the bottom 2 speakers per gender age-group pair with the least durations and include them in the zero-shot test set. For few-shot and many-shot splits, we uniformly sample utterances across genders, age groups, durations, and pitch in an attempt to ensure style diversity. Thus, this benchmark serves as a novel evaluation framework encompassing age and gender diversity to uniquely assess TTS performance across various demographics, ensuring robust generalization of models.

Table 4: Speaker counts and test durations of INDICVOICES-R Benchmark expanded across genders, age groups, and N-shot splits where ZS implies zero-shot (no training data for speaker), and $< K$ mins implies the speaker has less than $K$ minutes in the training data.

| Gender | Age Group | Speakers | | | | Durations (in hours) | | | |
|---|---|---|---|---|---|---|---|---|---|
| | | ZS | <5 mins | <10 mins | >10 mins | ZS | <5 mins | <10 mins | >10 mins |
| Male | 18-30 | 45 | 107 | 272 | 468 | 1.32 | 0.08 | 2.56 | 0.75 |
| | 30-45 | 44 | 75 | 177 | 308 | 1.72 | 0.06 | 2.13 | 0.95 |
| | 45-60 | 44 | 41 | 130 | 233 | 1.98 | 0.03 | 2.44 | 0.56 |
| | 60+ | 44 | 29 | 87 | 159 | 2.85 | 0.02 | 1.57 | 1.25 |
| Female | 18-30 | 46 | 130 | 258 | 409 | 1.57 | 0.08 | 2.23 | 1.08 |
| | 30-45 | 44 | 75 | 187 | 280 | 1.51 | 0.06 | 2.26 | 0.54 |
| | 45-60 | 44 | 59 | 135 | 178 | 2.21 | 0.05 | 2.03 | 0.92 |
| | 60+ | 41 | 25 | 78 | 91 | 2.41 | 0.03 | 1.66 | 0.58 |
| **Total** | | 352 | 541 | 1324 | 2126 | 15.55 | 0.41 | 16.88 | 6.63 |

# 5 Experiments with IndicVoices-R

We employ VoiceCraft [4], a Transformer-based architecture that employs a novel token re-arrangement and delayed stacking mechanism to causally predict audio codec tokens. Particularly, to study the utility of our dataset, we conduct two fine-tuning experiments starting from the publicly available 830M parameter checkpoint [4] trained on GigaSpeech. Specifically, we first fine-tune a model on the IndicTTS database and compare its performance on the benchmark against a model fine-tuned on INDICVOICES-R.

**Vocabulary Expansion** To facilitate the fine-tuning of the English pre-trained model on Indian languages, we extend the tokenizer and initialize the newly introduced token embeddings with a Gaussian distribution centered around $\mu_{old}$ with variance $\sigma_{old}^2$. The token embeddings are concatenated with the old embeddings. The vocab size was extended from 100 tokens to 1089 tokens pooled across all 22 languages. The tokenizer was extended to 1536 with a token padding multiple of 512 for training efficiency. We chose graphemes over phonemes for building our multilingual TTS system due to the lack of phonemizers for all Indian languages.

**Experimental Setup** We train VoiceCraft on 8X NVIDIA A100 40GB GPUs. We use the standard AdamW optimizer with $\beta_1 = 0.99$ and $\beta_2 = 0.999$, a weight decay of $\lambda = 10^{-2}$ and finetune it for over 50K steps. We use the Lambda learning rate schedule with an initial learning rate of $0.00001$, and dynamically batched with a maximum of 20,000 tokens per GPU.

**Results** We report N-MOS and speaker similarity (S-SIM) scores. To calculate speaker similarity, we present the cosine similarity between the embeddings of the ground-truth and synthesized samples, extracted from Wav2vec2 [45] fine-tuned on IV-R data using the IndicSUPERB pipeline [36]. As

---

[4]Pre-trained checkpoint: `https://huggingface.co/pyp1/VoiceCraft/blob/main/giga830M.pth`

observed in Table 5, VoiceCraft fine-tuned on INDICVOICES-R achieves better speaker similarity scores on the zero-shot benchmark with nearly on-par N-MOS compared to the model fine-tuned on IndicTTS.

Table 5: Zero-shot speaker evaluation of VoiceCraft, fine-tuned on IndicTTS and dataset, on the INDICVOICES-R Benchmark.

| Finetune dataset | NORESQA | S-SIM |
|------------------|---------|-------|
| IndicTTS | 3.83 | 78.93 |
| **IndicVoices-R (Ours)** | 3.64 | **88.18** |

## 6  Ethical Considerations and Limitations

We prioritize responsible data practices in developing IndicVoices-R (IV-R). The dataset leverages anonymized speech from its parent dataset IndicVoices and is released under the same CC BY 4.0 license. IndicVoices underwent rigorous ethical review and approval by the Institute Ethics Committee. It obtained explicit consent from each participant for the use of their speech data and ensured no offensive content was part of it. We acknowledge potential biases within the source data and encourage further exploration. Addressing the limitations of our work, we could not unlock the potential of conversational subset of IndicVoices due to their low sampling rates of $8\,\mathrm{kHz}$. Furthermore, although the dataset exhibits characteristics of natural speech, it has not attained the caliber of studio-recorded speech datasets. Henceforth, our commitment persists in the pursuit of continual enhancement and advancement.

## 7  Conclusion

In this work, we introduce IndicVoices-R, a first-of-its-kind large-scale multilingual TTS dataset encompassing 22 Indian languages. This dataset comprises 1704 hours of high-fidelity speech-text data, capturing a diverse range of scenarios and speaker demographics. IndicVoices-R facilitates research and development in zero-shot multilingual text-to-speech synthesis. To complement the dataset, we further introduce a comprehensive benchmark designed to evaluate the zero-shot, few-shot, and many-shot capabilities of TTS models on data across various age and gender groups, along with different text and speech styles. We empirically demonstrate the dataset's utility by training a competent zero-shot TTS system on IndicVoices-R. Our work paves the way for the development of more robust and speaker-agnostic TTS systems, ultimately fostering broader accessibility and inclusivity for Indian languages.

### Acknowledgments and Disclosure of Funding

This project was made possible through the dedicated efforts and collaboration of numerous organizations and participants. We extend our gratitude to Digital India Bhashini, the Ministry of Electronics and Information Technology of the Government of India, EkStep Foundation and Nilekani Philanthropies for their generous grant. We also express our sincere thanks to the Centre for Development of Advanced Computing, Pune (CDAC Pune[5]), for providing access to their PARAM-Siddhi supercomputer, which was instrumental in running our speech enhancement pipeline and model training. We also thank the entire team at AI4Bharat for their unwavering support and encouragement throughout our research journey.

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

# A  Appendix

## A.1  Datasheets for Datasets

The following section is answers to questions listed in datasheets for datasets

### A.1.1  Motivation

- **For what purpose was the dataset created?**
  INDICVOICES-R is created to scale-up Indian TTS to all 22 languages, covering a large number of speakers from various demographics. We also release INDICVOICES-R-Benchmark to evaluate the cross-speaker generalization of TTS systems for Indian voices spanning across age-groups and real life scenarios.

- **Who created the dataset (e.g., which team, research group) and on behalf of which entity (e.g., company, institution, organization)?**
  INDICVOICES-R is presented by AI4Bharat, a research lab at the Indian Institute of Technology-Madras, whose mission is to empower and elevate the potential of Indian languages in AI technologies by fostering open-source collaboration in datasets, models, and applications.

- **Who funded the creation of the dataset? If there is an associated grant, please provide the name of the grantor and the grant name and number.**
  The dataset creation was funded by Digital India Bhashini, the Ministry of Electronics and Information Technology of the Government of India, EkStep Foundation and Nilekani Philanthropies.

### A.1.2  Composition

- **What do the instances that comprise the dataset represent (e.g., documents, photos, people, countries**
  INDICVOICES-R contains speech-text pairs with additional metadata such as SNR, C50, speaking rate, gender, age-group, etc.

- **How many instances are there in total (of each type, if appropriate)?**
  INDICVOICES-R comprises a total a 1704.34 hours of speech-text pairs from approximately 690K utterances, containing 10,496 speakers across the 22 officially recognized languages of India. Please refer Table 3 for more details.

- **Does the dataset contain all possible instances or is it a sample (not necessarily random) of instances from a larger set?**
  Yes, this dataset is a complete set. Although the dataset is derived from IndicVoices, it is complete in itself and is intended to be used for TTS (Text-to-Speech) research and applications.

- **What data does each instance consist of?**
  The dataset contains speech-text pairs i.e., the filepath pointing to each audio file and the corresponding normalized text. Each instance also includes metadata on the speech from metrics like SNR, C50, pitch mean and speaking rate to metadata on the speaker like age-group, gender, etc.

- **Is there a label or target associated with each instance?**
  Yes, there are text, gender, speaker and language labels associated with each instance of the data.

- **Is any information missing from individual instances? If so, please provide a description, explaining why this information is missing (e.g., because it was unavailable). This does not include intentionally removed information, but might include, e.g., redacted text.**
  No.

- **Are relationships between individual instances made explicit (e.g., users' movie ratings, social network links)?**
  Yes, instances containing audio recordings from the same speaker are identified with the anonymized speaker ids, which is captured in the meta-data.

- **Are there recommended data splits (e.g., training, development/validation, testing)?**
  Yes, we provide the training and validation splits to ensure that there's no test-set leakage from the benchmark. We release this on our dataset webpage `https://github.com/AI4Bharat/IndicVoices-R`.

- **Are there any errors, sources of noise, or redundancies in the dataset?**
  No, the data does not have any errors, sources of noise or redundancies.

- **Is the dataset self-contained, or does it link to or otherwise rely on external resources (e.g., websites, tweets, other datasets)?**
  The dataset is self-contained and complete in itself.

- **Does the dataset contain data that might be considered confidential (e.g., data that is protected by legal privilege or by doctor–patient confidentiality, data that includes the content of individuals' non-public communications)?**
  No.

- **Does the dataset contain data that, if viewed directly, might be offensive, insulting, threatening, or might otherwise cause anxiety?**
  No.

- **Does the dataset relate to people**
  Yes, this dataset consists of audio recordings from multiple people but is derived from a source which has already ensured anonymization.

- **Does the dataset identify any subpopulations (e.g., by age, gender)?**
  Yes, the dataset provides desensitised metadata on age-groups, gender and more.

- **Is it possible to identify individuals (i.e., one or more natural persons), either directly or indirectly (i.e., in combination with other data) from the dataset?**
  No, the data has been thoroughly anonymized, ensuring that identification of individuals is not possible with the provided metadata.

- **Does the dataset contain data that might be considered sensitive in any way (e.g., data that reveals race or ethnic origins, sexual orientations, religious beliefs, political opinions or union memberships, or locations; financial or health data; biometric or genetic data; forms of government identification, such as social security numbers; criminal history)?**
  The data does not contain any sensitive metadata as such, but includes metadata from its parent source such as age-group, gender, occupation, and location, which have all been thoroughly desensitised.

### A.1.3 Collection Process

- **How was the data associated with each instance acquired?**
  We derive our dataset by enhancing speech-text pairs from the existing IndicVoices dataset, followed by careful filtering. The entire process has been discussed in Section 3 of the paper.

- **What mechanisms or procedures were used to collect the data (e.g., hardware apparatuses or sensors, manual human curation, software programs, software APIs)?**
  Our work transforms an existing dataset using several algorithms that have been discussed in Section 3.

- **If the dataset is a sample from a larger set, what was the sampling strategy (e.g., deterministic, probabilistic with specific sampling probabilities)?**
  N/A.

- **Who was involved in the data collection process (e.g., students, crowdworkers, contractors) and how were they compensated (e.g., how much were crowdworkers paid)?**
  This dataset was created by students and project staff who were paid standard industry rates.

- **Over what timeframe was the data collected?**
  N/A.

- **Were any ethical review processes conducted (e.g., by an institutional review board)?**
  An ethical review was conducted for the parent dataset which deemed the dataset to be fit for use for any downstream applications.

- **Does the dataset relate to people?**
  Yes, this dataset consists of audio recordings from multiple people but is derived from a source which has already ensured anonymization.

- **Did you collect the data from the individuals in question directly, or obtain it via third parties or other sources (e.g., websites)?**
  The parent data was acquired from IndicVoices (`https://ai4bharat.iitm.ac.in/indicvoices/`) which is a publicly released ASR dataset spanning across 22 languages.

- **Were the individuals in question notified about the data collection?**
  N/A.

- **Did the individuals in question consent to the collection and use of their data?**
  N/A. We re-purpose an existing dataset released under the CC-BY-4.0 license comprising of speech-text pairs for which explicit consent has already been collected.

- **If consent was obtained, were the consenting individuals provided with a mechanism to revoke their consent in the future or for certain uses?**
  N/A.

- **Has an analysis of the potential impact of the dataset and its use on data subjects (e.g., a data protection impact analysis) been conducted?**
  N/A.

### A.1.4 Preprocessing/cleaning/labeling

- **Was any preprocessing/cleaning/labeling of the data done (e.g., discretization or bucketing, tokenization, part-of-speech tagging, SIFT feature extraction, removal of instances, processing of missing values)?**

Our dataset is derived from an existing ASR dataset, and we thoroughly describe the steps of data preparation, filtering, and post-processing in Section 3.2.

- **Was the "raw" data saved in addition to the preprocessed/cleaned/labeled data (e.g., to support unanticipated future uses)?**
  N/A. The source of our parent dataset is publicly available.

- **Is the software that was used to preprocess/clean/label the data available?**

  1. **Preprocessing:** The data was made fit to be use for TTS systems by re-purposing the ASR data, IndicVoices through a cascade of systems:
     - HTDemucs (https://github.com/ZFTurbo/Music-Source-Separation-Training)
     - VoiceFixer (https://github.com/haoheliu/voicefixer)
     - DeepFilternet-3 (https://github.com/Rikorose/DeepFilterNet/)
  2. **Cleaning and Filtering:** Data stats such as C50, SNR, speaking rate, pitch stats, were computed using DataSpeech (`https://github.com/huggingface/dataspeech`) and NORESQA was calculated using the official implementation (`https://github.com/facebookresearch/Noresqa`).

### A.1.5 Uses

- **Has the dataset been used for any tasks already?**
  Yes, in this work we use INDICVOICES-R to train the first ever TTS model covering all 22 Indian languages and we report its performance scores on our proposed benchmark in Table 5.

- **Is there a repository that links to any or all papers or systems that use the dataset?**
  N/A.

- **What (other) tasks could the dataset be used for?**
  The dataset could be augmented and used for the several purposes its parent dataset boasts of including speaker diarization, speaker identification, speaker verification, language identification, intent detection, entity extraction, query by example, audio denoising, speaker re-construction, and speech separation.

- **Is there anything about the composition of the dataset or the way it was collected and preprocessed/cleaned/labeled that might impact future uses?**
  No.

- **Are there tasks for which the dataset should not be used?**
  No, but the authors call for responsible usage of the dataset.

### A.1.6 Distribution

- **Will the dataset be distributed to third parties outside of the entity (e.g., company, institution, organization) on behalf of which the dataset was created?**
  The dataset is publicly available for everyone to use under the said license.

- **How will the dataset will be distributed (e.g., tarball on website, API, GitHub)?**
  The dataset is released at `https://github.com/AI4Bharat/IndicVoices-R` and comprises of tarballs split across languages which can be downloaded.

- **When will the dataset be distributed?**
  13/06/2024 and onwards.

- **Will the dataset be distributed under a copyright or other intellectual property (IP) license, and/or under applicable terms of use (ToU)?**
  The dataset is released under CC-BY-4.0 License.

- **Have any third parties imposed IP-based or other restrictions on the data associated with the instances?**
  No.

- **Do any export controls or other regulatory restrictions apply to the dataset or to individual instances?**
  No.

### A.1.7 Maintenance

- **Who will be supporting/hosting/maintaining the dataset?**
  AI4Bharat, our research lab at IIT Madras, will support, host, and maintain this dataset.

- **How can the owner/curator/manager of the dataset be contacted (e.g., email address)?**
  Please contact the authors or the provider, AI4Bharat, through Github issues at `https://github.com/AI4Bharat/IndicVoices-R`.

- **Is there an erratum**
  No.

- **Will the dataset be updated (e.g., to correct labeling errors, add new instances, delete instances)?**
  Any updates or corrections if required will be reflected on the official GitHub page.

- **If the dataset relates to people, are there applicable limits on the retention of the data associated with the instances (e.g., were the individuals in question told that their data would be retained for a fixed period of time and then deleted)?**
  N/A.

- **Will older versions of the dataset continue to be supported/hosted/maintained?**
  We are currently at the first version of our dataset and intend to support all versions even during future releases.

- **If others want to extend/augment/build on/contribute to the dataset, is there a mechanism for them to do so?**
  We are open to discussing such requests and would kindly ask users to contact the authors for the same.

## A.2 Dataset Nutrition Label

| Variables | |
| --- | --- |
| *text* | Level 2 transcript. The text is normalized for abbreviations, numbers and standardized spellings. Please refer to IndicVoices for details on the normalization rules. |
| *lang* | Language of the utterance, listed in the ISO-639-1 format. |
| *samples* | Number of samples = Duration of audio $\times sampling\_rate$. |
| *verbatim* | Level 1 transcript. The text captures the spoken language as it is, without any standardization of spellings. |
| *normalized* | Level 2 transcript. The same as the field *text*. |
| *speaker_id* | The ID of the speaker in the utterance. |
| *scenario* | The speech may be Read-Speech or Extempore. In case of Extempore, the speakers were not provided with a script, instead they had to answer according to a prompt or question. |
| *task_name* | Represents the domain to which the utterance belongs. The dataset covers more than 70 domains, ranging from styles like Alexa commands to topics like Games, Tourism and Technology. |
| *gender* | The participants were asked to indicate their gender, with the options 'Male', 'Female' and 'Others'. |
| *age_group* | The data has speakers from four age groups, 18-30 years, 30-45 years, 45-60 years and 60+ years. |
| *job_type* | Participants are classified into four job categories - 'Student', 'Unemployed', 'Blue Collar', 'White Collar'. |
| *qualification* | Participants could choose from four qualification types - 'No Schooling', 'Post-Grad + PhD', 'Upto 12th', 'Undergrad and Grad'. |
| *area* | The place of recording - Rural or Urban. |
| *district* | The district within the state in which the utterance was recorded. |
| *state* | The state of India in which the utterance was recorded. |
| *occupation* | Participants were requested to declare their occupation. Our dataset covers utterances from all walks of life. |
| *filename* | Name of the audio file corresponding to the utterance. |
| *duration* | Duration of the utterance, reported in seconds. |
| *cer* | This is the Character Error Rate between Level 1 and Level 2 transcripts. |
| snr | The Signal-to-Noise ratio calculated using Brouhaha. |
| C50 | The reverberation score, C50, calculated using Brouhaha. |
| utterance_pitch_mean | The mean of the pitch within the utterance computed using PENN. |
| utterance_pitch_std | The standard deviation of the pitch within the utterance computed using PENN. |

## A.3 Qualitative Examples from the INDICVOICES-R Benchmark

| Language Code | Scenario | Sentences |
|---|---|---|
| asm | Alexa Commands | **Text:** এটা ষ্টেটোছ দিয়া এইটো কৈ মই প্ৰমোচন পাইছলোঁ |
| | | **Transliteration:** atta status diya eitu koi moi promuson paisu |
| | | **Translation:** Put a status saying 'I got a promotion'. |
| ben | Domain of Interest - Agriculture | **Text:** তাতে একটু সময় বাঁচলো মানুষের আর অল্প সময়ের মধ্যে অনেক জমি জায়গা চাষবাস করা হতো |
| | | **Transliteration:** tate ektu somomo banchlo manusher aar alpo somepor modhye onek jommy jagiba chashbas koraa hoto |
| | | **Translation:** This saved time for the people and in a short time, a lot of land was cultivated. |
| brx | Umang Commands | **Text:** केराला रायजोआव स्पुटनिक गोनां गासैबो सेन्टारफोरखौ आंनो दिन्थि |
| | | **Transliteration:** kerala raijwao spootnik gwnang gaswibw centrepwrkwo angnw dinti |
| | | **Translation:** Show me all centres in state Kerala having Sputnik |
| doi | Domain of Interest - Animal Husbandry | **Text:** डेरी आह्ले जेह्ड़े फार्मर होंदे दुद्ध शुद्ध आह्ले ते ओह् ते गमां ते मंजां पालदे |
| | | **Transliteration:** dairy ahle jehde former honde duddh shuddh ahle the oh the gamaan the manjaan paalade |
| | | **Translation:** Farmers like the dairy farmers would raise milk and they would raise cows and cattle |
| guj | Bigbasket Commands | **Text:** કેસરી સેફ્રોનને પરત કરવાની શું પ્રક પ્રક્રિયા છે બરાબર |
| | | **Transliteration:** kesari sephrone parat karvaanee shun prak prakriyaa chhe barabar |
| | | **Translation:** What is the procedure to return Kesari Saffron? |
| hin | Domain of Interest - Business | **Text:** बहुत सारे ऐसे बिजनेस हैं जैसे कि कपड़ों का बिजनेस है |
| | | **Transliteration:** bahut saare aise business hain jaise kii cups kaa business hai |
| | | **Translation:** There are many such businesses like clothing business. |
| kan | Know your participant - Traveling | **Text:** ಎಲ್ಲರು ಮಾಸ್ಕ್ ಳನ್ನು ಕಡ್ಡಾಯ ಹಾಕಬೇಕಂತಿತ್ತು ಮತ್ತೆ |
| | | **Transliteration:** ellaru maskgalannu kaddaya haakabekantittu matte |
| | | **Translation:** It was mandatory for everyone to wear masks again |
| kas | Daily Life | **Text:** وَنکِنَس چھُ کِچھنَس مَنٛز یہِ چھُ ہیٖٹَر چھُ بُخأٚری چھُ |
| | | **Transliteration:** venkenas chu kichnas menz yeh chu heater chu bukheyr chu |
| | | **Translation:** Currently in the kitchen, it is hotter than usual. |
| kok | Coordinator Section | **Text:** तुमच्या मते निताळसाणीचें कितें म्हत्व आसा |
| | | **Transliteration:** tumchya matte nitalsaanichen kitein mhatva aasaa |
| | | **Translation:** What do you think is the importance of cleanliness? |
| mai | Keywords Spotting | **Text:** असहमति |
| | | **Transliteration:** asahamati |
| | | **Translation:** Disagreements |
| mal | Domain of Interest - Health | **Text:** ഞങ്ങളുടെ ആരോഗ്യകേന്ദ്രമെന്ന് പറയുമ്പോൾ ഇവിടെ ഇവിടെ ഹോസ്പിറ്റലുണ്ട് ജില്ലാ ആശുപത്രി |
| | | **Transliteration:** njangalude aarogyakendramennu parayumbol ivide ivide hospitalundu jilla ashupatri |
| | | **Translation:** When we say our health center, there is a hospital here, the district hospital |

| | | |
|---|---|---|
| mni | Domain of Interest - Culture | **Text:** ꯄꯤꯗꯨꯅ ꯇꯦꯂꯥꯇꯤ

**Transliteration:** piduna laakli

**Translation:** It is being supplied. |
| mar | Domain of Interest - Entertainment | **Text:** राजकारणामध्ये सुद्धा ते भाग घेतात समाजसेवा करतात असा असा वेळ तो आपला फावल्या वेळामधे आपला करतात आणि ये

**Transliteration:** rajkaranamadhye suddha they bhag ghetat samajseva karatat assa assa vela too aapala favalya velamadhe aapala karatat aanee yeey

**Translation:** He also participates in politics and does social service in his spare time. |
| nep | Domain of Interest - Animal Husbandry | **Text:** के रे दुधहरू दुहुनु जाँदाखेरि कति गाईले त लात हान्ने गर्छ कति गाईले

**Transliteration:** key ray dudhaharu duhunu jandakheri kathi gaile tata lat hanne garchha kathi gaile

**Translation:** How many cows kick while going to milk milk? |
| ori | Domain of Interest - Legal | **Text:** ତେଣୁ ଯଦି ଶିଶୁମାନେ ଶ୍ରମିକ ହୋଇଯିବେ ତାହେଲେ ଦେଶ ବୁଡ଼ିଯିବ

**Transliteration:** tenu jadi shishumane shramika hoijibe taahele desh bujiba

**Translation:** So if children become labourers, the country will sink. |
| pan | Domain of Interest - Health | **Text:** ਪੰਜਾਬ ਦੇ ਪ੍ਰਸ਼ਾਸਨ ਵੱਲੋਂ ਸਹਿਤ ਸੰਭਾਲ ਉੱਤੇ ਬਹੁਤ ਖਾਸ

**Transliteration:** punjab dey parshasana vallon sihat sambhal utte bahut khaas

**Translation:** Punjab's administration is very particular on healthcare |
| san | Domain of Interest - Religion | **Text:** विघ्ननाशाय गणहोमं वा ऐश्वर्य ऐश्वर्यादिप्राप्त्यर्थं लक्ष्मी उपासना वा

**Transliteration:** vighnanaashaaya ganahomam wa aishvarya aishwaryaadipraaptyartham lakshmi upasana wa

**Translation:** Vighnashay Gana Homam or Lakshmi Puja for wealth and prosperity. |
| sat | Domain of Interest - Geography | **Text:** ᱡᱟᱣᱠᱟ. ᱥᱮ ᱠᱟᱨᱟᱢᱤ ᱯᱮ ᱜᱟᱨᱟᱢ-ᱠ ᱟᱢ ᱩᱸᱵᱟᱢ. ᱞᱟᱜᱟᱭ ᱚᱥ ᱠᱟᱨᱟᱢᱤ ᱯᱟᱨᱵᱚᱱ ᱩᱯᱟᱥᱚᱜ-ᱠ

**Translation:** Karam is performed during the night and Karam is performed along with the puja. |
| snd | Digital Payment Commands | **Text:** छा तव्हां जिओ पे तां मुंहिंजो नैनीताल बैंक मास्टरकार्ड हटाए सघंदा आहियो

**Transliteration:** chhaa tavhaan geo pay taan munhinjo nainital bank mastercard hataaye saghanda aahiyo

**Translation:** Can you remove my Nainital Bank Mastercard from Jio Payments Bank? |
| tam | District Specific | **Text:** இந்த மதிரிஅழகர் திலகர் திடல்

**Transliteration:** inththa madhiri azhagar thilagar thidal

**Translation:** This is the type of beautiful tilak. |
| tel | Know your participant - Basic | **Text:** సొంతూరు అంటే చాలా ఇష్టం అన్నమాట

**Transliteration:** sonturu ante chaalaa ishtam annamata

**Translation:** I love being at home |
| urd | District Specific | **Text:** اک چھوٹے بچوں کا

**Transliteration:** chhote bachon kaa

**Translation:** the little children's |

### A.4 Authors' Statement, Ethics, and Privacy

In this work, we introduce a transformed dataset derived from the previously publicly released INDICVOICES dataset. We re-release our transformed dataset under the same CC-BY-4.0 license to ensure its continued accessibility and utility for the research community. The parent dataset's data collection process underwent a rigorous ethical review and approval by the Institute Ethics Committee, ensuring comprehensive ethical standards were upheld. This included providing all instructions in participants' native languages, fully informing participants about the purpose of data collection, obtaining explicit consent, and offering appropriate compensation. We have adhered to the same ethical standards in our work, safeguarding the privacy and confidentiality of the participant's personal identifiable information (PII), and ensuring the data remains anonymized and protects sensitive information. All personnel involved in the transformation and handling of the dataset were appropriately compensated, maintaining ethical standards in labor practices. By releasing this transformed dataset under the CC-BY-4.0 license, we support its free and open use, including for commercial purposes, in line with the original dataset's licensing terms. We commit to updating our dataset in accordance with any changes to the license terms of the original dataset, ensuring continued compliance and ethical integrity.

### A.5 DOI

The Digital Object Identifier for the IndicVoices-R is 10.5281/zenodo.14016558.

