# OpenReview forum: "IndicVoices-R: Unlocking a Massive Multilingual Multi-speaker Speech Corpus for Scaling Indian TTS"
_NeurIPS.cc/2024/Datasets_and_Benchmarks_Track — NeurIPS 2024 Track Datasets and Benchmarks Poster_

### Official Review · Reviewer_37Ua · 2024-07-08
**Interesting work but lack of some comparisons**

**Rating:** 4
**Confidence:** 5
**Correctness:** The claims are correct. The dataset i…

**Review:**

A refined version of exising IndicVoices was presented, which provides a large-scale dataset for TTS in Indian voice.
The establish of the dataset seems OK.
However, the paper mainly doing some data cleaning and speech enhancement to IndicVoices, which contributes quite little.

**Strengths:**

The paper provided a way to bulid up large-scale TTS dataset.

**Additional Feedback:**

They paper could be improved if the authors analysis more on the data.

**Clarity:**

The paper is generally well-written. The authors have clearly articulated their methodology and experiments.

**Documentation:**

Yes

**Ethics:**

No, I don't.

**Limitations:**

Yes they have addressed some of the limitations.

**Opportunities For Improvement:**

The dataset is built on speech denoising and enhancement, and they chose HTDemucs and VoiceFixer for denosing and dereverbration. Why did they choose them? Did the authors has a comparison on the audio quality between the processed data and unprocessed data?

The idea is much similar to the LibriTTS-R paper. Did the authors compare with that paper on the method?

The proposed zero-shot and many-shot TTS methods did not have any novelity. They are exising method and just finetune in the new data.

**Relation To Prior Work:**

The proposed dataset is totally based on an existing dataset, i.e. Indicvoices. The method to build the dataset is similar to LibriTTS-R, but the paper did not compare with the method in LibriTTS-R.

**Summary And Contributions:**

This paper enhances existing large-scale ASR datasets containing natural conversations collected in low-quality environments to generate high-quality TTS training data. The authors also use this dataset to train a TTS model for Indian voices.

---

> ### Author Rebuttal · Authors · 2024-08-17
>
> We thank the reviewer for the feedback and suggestions, and would like to address the points raised -
>
> **Point 1.** *The dataset is built on speech denoising and enhancement, and they chose HTDemucs and VoiceFixer for denosing and dereverbration. Why did they choose them? Did the authors has a comparison on the audio quality between the processed data and unprocessed data?*
> * HTDemucs and VoiceFixer were selected for denoising and dereverberation due to their demonstrated effectiveness and widespread adoption in the speech processing community. These models have undergone rigorous evaluation and have proven to be reliable tools for improving audio quality.
> * Regarding the impact of the denoising pipeline on audio quality, we had deferred this as the improvement in quality was perceptually very evident. However, we understand the importance of quantifying this information and will provide a detailed analysis and comparisons between processed and unprocessed audio samples in the appendix.
>
>
>
> **Point 2.** *The idea is much similar to the LibriTTS-R paper. Did the authors compare with that paper on the method?*
> * While our work shares a similar goal with LibriTTS-R of improving speech quality, our approach differs significantly. LibriTTS-R employs a text-informed parametric resynthesis method, which is not publicly available due to the lack of open-sourced code and models. In contrast, we propose a language-agnostic denoising and restoration pipeline that is computationally efficient and can be applied without retraining for different languages.
>
>
> **Point 3.** *The proposed zero-shot and many-shot TTS methods did not have any novelity. They are exising method and just finetune in the new data.*
> * Our primary contribution lies in the creation and meticulous curation of a dataset specifically designed for training and evaluating robust TTS systems for the Indian languages. The NeurIPS Datasets and Benchmarks track (https://neurips.cc/Conferences/2024/CallForDatasetsBenchmarks) accepts submissions related to “New datasets, or carefully and thoughtfully designed (collections of) datasets based on previously available data”.
> * The zero-shot and many-shot experiments serve as a preliminary demonstration of the dataset's potential to facilitate advancements in TTS technology for Indian languages. We believe that this dataset will serve as a strong foundation for future research and development in this area, enabling researchers to explore novel methods and techniques for improving TTS quality.

---

> > ### Author Response · Authors · 2024-09-01
> > **Official comment from the Authors**
> >
> > As a reminder, the discussion period is coming to a close soon. We encourage you to share any additional thoughts, questions, or comments you may have. Your insights will be invaluable in helping us improve our work.
> >
> > Thank you for your time and contributions.

---

### Official Review · Reviewer_2UXi · 2024-07-20
**An interesting large-scale TTS corpus with promising and scalabale corpus collection method for TTS**

**Rating:** 7
**Confidence:** 4
**Correctness:** The dataset construction is correct a…
**Clarity:** The paper is clearly written

**Review:**

The paper presented a novel method to collect large-scale TTS corpus from ASR which has a comparable quality to existing natural TTS corpus introducing IndicVoices-R, the largest TTS corpus for languages. The size of the corpus is massive, consisting of 1704 hours with 10496 speakers covering 22 languages. The quality of IndicVoices-R is compared with other existing Indic TTS corpora by comparing their SNR, C50, F0, and NORESQA-MOS. The paper provides a great details regarding the IndicVoices-R corpus and benchmark, along with diversity comparison with existing corpora. Nonetheless, the empirical comparison is rather limited as it is only compared with model trained on IndicTTS tested on the IndicVoices-R benchmark, and only the aggregated results are reported. It will be great if the comparison is presented per language as the language coverage in IndicTTS is different than the one in the IndicVoices-R.

**Strengths:**

- The paper introduces a novel approach for restoring Indic ASR corpus into a comparable quality of Indic TTS data.
- The paper introduces IndicVoices-R, the largest and most diverse Indic TTS corpus to date.
- The paper is well-written with many insightful analysis about the corpora comparison.

**Additional Feedback:**

- Figure 5 is using "ISO-639-2" or "ISO-639-3" instead of "ISO-639-1" as it a three-letter code

**Documentation:**

Yes, a sufficient details is provided regarding the dataset collection, availability, and responsible use

**Ethics:**

-

**Limitations:**

Yes, limitation is adequately addressed in the paper.

**Opportunities For Improvement:**

- It will be interesting if there is an experiment on the VoiceCraft using the original IndicVoices data to see the effect of restoration pipeline to the resulting TTS model.
- Since IndicTTS only covers 14 languages, it will be interesting if the per language comparison of IndicVoice-R and IndicTTS fine-tuned model is presented in the paper.

**Relation To Prior Work:**

The paper show a great detail of comparison with prior works

**Summary And Contributions:**

The paper presented a novel method to collect large-scale TTS corpus from ASR which has a comparable quality to existing natural TTS corpus introducing IndicVoices-R, the largest TTS corpus for languages to date.

---

> ### Author Rebuttal · Authors · 2024-08-17
>
> We thank the reviewer for the feedback and suggestions, and would like to address the points raised -
>
> **Point 1.** *It will be interesting if there is an experiment on the VoiceCraft using the original IndicVoices data to see the effect of restoration pipeline to the resulting TTS model.*
>
> * We appreciate the suggestion to experiment with VoiceCraft using the original IndicVoices dataset. However, the original IndicVoices was primarily constructed to train ASR models and make them robust to noisy conditions. In contrast, TTS models require high-quality, clean audio data to generate natural-sounding speech. The presence of background noise in IndicVoices would degrade the performance of a TTS model trained on it.
> * Given the computational constraints required to train an 830M parameter model on over 1500 hours of noisy data for this experiment, we adopt a more targeted approach of data cleaning and denoising. We encourage the community to explore this direction further.
> * To provide a clearer understanding of the challenges involved, we have made available audio samples of the original IndicVoices and the restored version at https://ai4bharat.iitm.ac.in/indicvoices_r/.
> * Additionally, we provide a few plots to compare the speech quality of the original IndicVoices against our restored version and would be glad to include it in the appendix - [C50](https://github.com/AI4Bharat/IndicVoices-R/blob/master/assets/c50_violin_plot.png), [SNR](https://github.com/AI4Bharat/IndicVoices-R/blob/master/assets/snr_violin_plot.png) and [N-MOS](https://github.com/AI4Bharat/IndicVoices-R/blob/master/assets/pesq_histogram_comparison.png).
>
>
>
> **Point 2.** *Since IndicTTS only covers 14 languages, it will be interesting if the per language comparison of IndicVoices-R and IndicTTS fine-tuned model is presented in the paper.*
> * We appreciate the suggestion to provide a more granular analysis of per-language performance. While we present consolidated scores across all the 14 languages in Table 5 for brevity, we understand the value of language-specific insights. We will include a detailed per-language comparison of IndicVoice-R and IndicTTS fine-tuned models in the appendix.
>
>
> **Point 3.** *Figure 5 is using "ISO-639-2" or "ISO-639-3" instead of "ISO-639-1" as it a three-letter code*
> * We are extremely grateful to the reviewer for carefully scrutinizing our work. We use ISO-639-3 codes and will update the caption accordingly.

---

> > ### Comment · Reviewer_2UXi · 2024-08-21
> >
> > Thank you authors for the response.
> >
> > The response addresses some of my concerns especially on the comparison of IndicVoices and IndicVoices-R. There seems to be a significant improvement on SNR based on the provided figures. It will be great, if these details can be also included in the main content.
> >
> > Overall, I feel more confident with the paper and I will increase the score accordingly.

---

> > > ### Author Response · Authors · 2024-08-21
> > > **Official Comment by Authors**
> > >
> > > Thank you for your valuable feedback. We're glad the response addresses your concerns. We willl include these details in the manuscript. We deeply appreciate your support and consideration in adjusting the score.

---

### Official Review · Reviewer_Aehz · 2024-07-24
**A well-curated, relevant and useful dataset for the community.**

**Rating:** 7
**Confidence:** 5
**Correctness:** Yes, the dataset construction and its…
**Clarity:** Yes

**Review:**

- Clarity: The paper is generally well written and the dataset curation pipeline is easy to follow.

- Quality / Significance: The dataset is well-curated with an interesting data pipeline to filter out data from IndicVoices dataset. The authors use several quality metrics (C50, SNR, speaking rates) to extract and enhance the speeches from IndicVoices. I strongly believe that the curated dataset IV-R would be quite useful for the community as it opens up TTS tasks on 22 other Indian languages. The high-quality speech-text data for these languages is hardly available publicly.

- Originality: The dataset contribution seems to be a pioneering work for TTS tasks on 22 Indian languages.

- Cons: Notable limitation of the work is lack of relevant experiments to show the strengths of the dataset. Though the authors shows zero-shot speaker evaluation of IV-R, it would be good to run the experiments on different settings of TTS (few-shot, many shot, etc). It would be nice to show TTS generalization across genders and age groups.

**Strengths:**

- A large-scale speech-text data collection for TTS task for 22 Indian languages. The dataset comprises 1704 hours of high-fidelity speech-text data from 10,964 speakers, capturing a diverse range of scenarios and speaker demographics.

- The authors proposed a sensible data curation pipeline with preprocessing, demixing, dereverberation, and filtering techniques to fetch enhanced speeches from IndicVoices. Further, the authors shows several quality metrics like C-50 for reverberation, N-MOS, SNR to compare IV-R dataset with other TTS datasets. The authors also shows several statistics to show the speaker diversity and quality of speeches of IV-R.

**Additional Feedback:**

Comments are provided in the previous sections.

**Documentation:**

The dataset is well documented and available online for public review. The authors also provide data sheet and FAQs for the dataset and licensing details.

**Ethics:**

No. Though human subjects are involved, the paper acknowledges that they took consent from the subjects for the data release.

**Limitations:**

Yes, the authors have discussed the limitations and acknowledges the potential negative societal impact.

**Opportunities For Improvement:**

Limitations of the work is discussed in Review section. Some minor pointers:

- Provide split of train-val-test of the dataset for different settings (zero-shot, few-shot, many-shot)
- It would be good to pick few set of baselines to benchmark different tasks.

**Relation To Prior Work:**

Prior work comparison is addressed clearly in the paper.

**Summary And Contributions:**

- The work introduces IndicVoices-R (IV-R), a large-scale multilingual TTS dataset comprising of 22 Indian languages. This dataset comprises 1704 hours of high-fidelity speech-text data from 10,964 speakers, capturing a diverse range of scenarios and speaker demographics.
- The authors shows several metrics like C-50 for reverberation, N-MOS, SNR to display the quality of IV-R with gold-standard 12 TTS datasets like LJSpeech, LibriTTS, and IndicTTS. The authors also shows several statistics to show the speaker diversity and quality of speeches of IV-R.
- Finally, the work demonstrates that fine-tuning an English pre-trained model on a combined dataset of high-quality IndicTTS and IV-R dataset results in better zero-shot speaker generalization compared to fine-tuning on the IndicTTS dataset alone.

---

> ### Author Rebuttal · Authors · 2024-08-17
>
> We thank the reviewer for the feedback and suggestions, and would like to address the points raised -
>
> **Point 1.** *Notable limitation of the work is lack of relevant experiments to show the strengths of the dataset. Though the authors shows zero-shot speaker evaluation of IV-R, it would be good to run the experiments on different settings of TTS (few-shot, many shot, etc). It would be nice to show TTS generalization across genders and age groups.*
>
> * Thank you for your insightful feedback. We understand the importance of conducting experiments under various TTS settings to demonstrate the dataset's strengths. However, we believe that the zero-shot speaker evaluation of IV-R already provides a strong indication of the dataset's utility. Zero-shot scenarios are particularly challenging and serve as a rigorous test of a model's generalization capabilities. The successful performance of our dataset in this demanding context highlights its robustness and versatility. While additional experiments could offer further insights, the zero-shot results alone demonstrate the dataset's capacity to generalize effectively, underscoring its value for a wide range of applications.
>
> **Point 2.** *Provide split of train-val-test of the dataset for different settings (zero-shot, few-shot, many-shot)*
>
> * We release the train-test splits on the official dataset page which can be found here - https://ai4bharat.iitm.ac.in/indicvoices_r/
> The file structure is as follow -
> ```
> ├── Assamese
> │   ├── gt10
> │   │   └── metadata_test.json
> │   ├── lt10
> │   │   └── metadata_test.json
> │   ├── lt5
> │   │   └── metadata_test.json
> │   ├── metadata_test.json
> │   ├── metadata_train.json
> │   └── wavs
> ├── Bengali
> │   ├── gt10
> │   │   └── metadata_test.json
> │   ├── lt10
> │   │   └── metadata_test.json
> │   ├── lt5
> │   │   └── metadata_test.json
> │   ├── metadata_test.json
> │   ├── metadata_train.json
> │   └── wavs
> ```
>
> **Point 3.** *It would be good to pick few set of baselines to benchmark different tasks.*
> * We release the first baseline for zero-shot speaker dataset for Indian languages. No open-source zero-shot TTS model for Indian languages exists prior to this work, to the best of our knowledge. We will be happy to evaluate and post a leaderboard for newer models once they’re released.

---

> > ### Comment · Reviewer_Aehz · 2024-08-21
> >
> > Thanking the authors for detailed response.
> >
> > Overall, I feel good with the paper and keep my ratings unchanged.

---

> > > ### Author Response · Authors · 2024-08-22
> > > **Official comment by Authors**
> > >
> > > Thank you for taking the time to review our manuscript and for your positive feedback. We appreciate your thoughtful consideration and are glad that our responses addressed your concerns.

---

### Official Review · Reviewer_8q3k · 2024-07-25

**Rating:** 5
**Confidence:** 4
**Correctness:** Fair.
**Clarity:** Need to be improved.

**Review:**

IndicVoices-R represents a new version in the availability of better diverse, and large-scale TTS data for Indian languages.  IndicVoices included both read-speech and conversational formats, providing diverse speaker demographics and speaking styles. This diversity is used for achieving zero-shot speaker evaluation in TTS models. A filtering process ensured that only high-quality audio samples were included in the final dataset, based on metrics such as C50, SNR, pitch mean, and speaking rate.

By addressing the limitations of the original ASR dataset and introducing some enhancement processes, the authors have created a resource for developing robust and inclusive TTS systems for Indian languages.

**Strengths:**

- IV-R is presented as the largest TTS dataset for Indian languages, encompassing all 22 scheduled languages of India.
- The authors applied advanced denoising, dereverberation, and speech enhancement techniques to improve the quality of the dataset.
- The IV-R Benchmark is introduced to evaluate the zero-shot, few-shot, and many-shot speaker generalization capabilities of TTS models.

**Additional Feedback:**

The reliance on subjective metrics like N-MOS for evaluating speech quality might not capture all TTS performance, such as prosody, intonation, and naturalness in various speaking styles.

**Documentation:**

Need to be improved.

**Ethics:**

No.

**Limitations:**

Related to its parent sent, the authors are recommended to continually review and update ethical guidelines to address emerging concerns in data privacy and bias mitigation

**Opportunities For Improvement:**

- The exclusion of 8kHz recordings, while necessary for quality, means potentially useful data is discarded. Future advancements in enhancement techniques might enable the use of such lower-quality recordings.
- The advanced enhancement processes like demixing, dereverberation, and filtering are computationally intensive, which degrades some information during audio processing.

**Relation To Prior Work:**

Some connections.

**Summary And Contributions:**

The paper introduces IndicVoices-R (IV-R), a large-scale multilingual multi-speaker speech corpus for Indian languages designed to address the lack of high-quality TTS (text-to-speech) data for Indian languages. The dataset comprises 1,704 hours of speech from 10,496 speakers across 22 Indian languages. The dataset was derived from an existing ASR (automatic speech recognition) dataset, IndicVoices, and was enhanced using various denoising and speech enhancement techniques.

However, the reference format and paper outline have some issues. e.g., please use footnote for the website link and use [1,2] instead of [1;2] in the citation format.

---

> ### Author Rebuttal · Authors · 2024-08-17
>
> We thank the reviewer for the feedback and suggestions, and would like to address the points raised -
>
> **Point 1.** *However, the reference format and paper outline have some issues. e.g., please use footnote for the website link and use [1,2] instead of [1;2] in the citation format.*
> * We thank the reviewer for their careful attention to detail. We appreciate the feedback regarding the reference format and paper outline. We will incorporate footnotes for website links and adjust the citation format to [1, 2] as suggested.
>
>
> **Point 2.** *The exclusion of 8kHz recordings, while necessary for quality, means potentially useful data is discarded. Future advancements in enhancement techniques might enable the use of such lower-quality recordings.*
> * We appreciate the reviewer's point regarding the trade-off between data quantity and quality. Our approach successfully restored 83% of the original data, which we believe strikes a balance between maintaining a substantial dataset size and ensuring data quality. The scarcity of paired telephony and full-band speech datasets in Indian languages presents considerable challenges in developing robust speech enhancers. Recognizing the potential value of this data, we are actively investigating advanced techniques to address the complexities involved in narrowband to fullband conversion. Our ongoing research aims to further improve both the quantity and quality of usable data for Indian language speech enhancement tasks.
>
> **Point 3.** *The advanced enhancement processes like demixing, dereverberation, and filtering are computationally intensive, which degrades some information during audio processing.*
> * We acknowledge the potential for information loss during enhancement. As detailed in Section 3.2, Step 5, we carefully consider data filtration and quality control and minimize sub-par samples that cannot be used for TTS.
> * Furthermore, Section 3.3 provides a comprehensive comparison to existing TTS datasets, demonstrating the overall quality of our processed data. Additionally, we provide a few plots to compare the speech quality of the original IndicVoices against our restored version and would be glad to include it in the appendix - [C50](https://github.com/AI4Bharat/IndicVoices-R/blob/master/assets/c50_violin_plot.png), [SNR](https://github.com/AI4Bharat/IndicVoices-R/blob/master/assets/snr_violin_plot.png) and [N-MOS](https://github.com/AI4Bharat/IndicVoices-R/blob/master/assets/pesq_histogram_comparison.png).
>
> **Point 4.** *The reliance on subjective metrics like N-MOS for evaluating speech quality might not capture all TTS performance, such as prosody, intonation, and naturalness in various speaking styles.*
> * N-MOS [REF] is a robust metric for evaluating TTS quality, even in out-of-distribution scenarios, demonstrating a stronger correlation with human judgments than other automatic MOS metrics. While we acknowledge the limitations of relying solely on objective measures, N-MOS provides a valuable initial assessment of overall quality.
> * To complement N-MOS, we report additional metrics such as pitch mean and standard deviation as proxies for prosody and intonation. These metrics offer insights into specific speech attributes that are not captured by overall quality scores.
> * We understand the importance of human subjective evaluation. However, the prohibitive cost of conducting large-scale MOS or MUSHRA tests on datasets of this scale renders them impractical. To facilitate qualitative assessment, we have made audio samples publicly available at https://ai4bharat.iitm.ac.in/indicvoices_r/ .
>
> [REF] Manocha, P., & Kumar, A. (2022). Speech Quality Assessment through MOS using Non-Matching References. Interspeech.

---

> > ### Comment · Reviewer_8q3k · 2024-08-27
> >
> > I have read the authors' response. It is a borderline paper to me since
> >
> > - the data is a version of IndicVoices subset and
> > - related analysis on Indian geography dynamic is relatively shallow.
> >
> > I will keep my original score and more on the rejection side.

---

### Author Response · Authors · 2024-08-24
**Official comment by Authors**

Dear Reviewers,

Thank you for taking the time to review our paper. We appreciate your thoughtful feedback and insights.

Please feel free to raise any concerns or questions you may have, no matter how minor they seem. Your feedback will help us improve our paper significantly.

We are committed to addressing your comments and ensuring that our paper meets the highest standards of quality.

---

### Decision · Program_Chairs · 2024-09-26

**Decision:**

Accept (Poster)

**Comment:**

This paper introduces IndicVoices-R, a multilingual multi-speaker speech corpus containing 1704 hours of speech for all the 22 official Indian languages. The dataset was derived from an existing IndicVoices and re-purposed for TTS by extensive preprocessing and cleaning such as (1) only selecting higher frequency audios (i.e 44kHz rather than 8kHz) (2) reducing background noise. (3) speech enhancement, and (4) additional filtering to improve the audio quality.

Overall reviewers see that this is a good contribution especially since it coves all Indian official languages. There are few concerns about the data enhancement process such as the discarding of recordings with 8kHz, the computational intensive nature of the preprocessing, and little contribution since it the major contribution is data cleaning. However, I think this is still a very good contribution for several Indian languages, being the largest single TTS resource for these low-resource languages.